# Behaviour and welfare assessment of autochthonous slow-growing rabbits: The role of housing systems

**Laura Ozella, Stefano Sartore, Elisabetta Macchi, Isabella Manenti⊙, Silvia Mioletti, Barbara Miniscalco, Riccardo Crosetto, Patrizia Ponzio, Edoardo Fiorilla⊙***,
**Cecilia Mugnai⊙**

Department of Veterinary Sciences, University of Turin, Grugliasco, TO, Italy

\* edoardo.fiorilla@unito.it

**Data Availability Statement:** All relevant data is available at the public repository Zenodo, https://doi.org/10.5281/zenodo.10979248.

## Abstract

Understanding the farming system impact on animals is crucial for evaluating welfare. Rabbits exhibit distinct behaviours influenced by their surroundings. The conditions in which they are raised directly influence behaviour and stress responses, emphasizing the importance of providing an optimal environment for their overall well-being and growth. In this study, we assessed the behaviour and welfare of two Italian local rabbit populations, namely the grey rabbit of Carmagnola and the grey rabbit of Monferrato. These rabbits are not yet officially recognized as breeds, but they are commonly used in Italy for meat production and represent a distinctive phenotype and local heritage among farmers and consumers. We analysed the behavioural patterns, physiological responses, and blood parameters of the animals to assess the influence of both age and three distinct housing systems (traditional single cages, group farming, and a mixed system) on rabbits' welfare. In this study, 294 weaned males with 35 days old were divided into three housing systems with seven replicates each until reaching slaughtering age (100 days of age). A traditional single cage system, a group farming with 10 animals each replicate and a Mixed pilot system with 10 rabbits initially grouped, then transferred to single cages. The findings from the behavioural analysis and the evaluation of salivary and hair corticosterone levels demonstrate that both the housing system and the age of the rabbits exerted significant effects on their welfare. Rabbits in group housing displayed a wider range of behavioural patterns, including increased kinetic activities such as running, walking, and exploration. However, this housing system was associated with higher levels of both salivary and hair corticosterone, indicating a high acute and chronic stress condition. The single cage system was associated with higher levels of acute stress and a low frequency of kinetic activities and social interactions, with a predominant behaviour of turning on themselves. The age factor significantly influenced the occurrence of behaviours, with younger rabbits exhibiting higher levels of kinetic activities, while social behaviours such as attacks and dominance were more prevalent as the rabbits reached sexual maturity (around 80–85 days of age). Moreover, the attainment of sexual maturity coincided with an increase in salivary corticosterone levels. We found a significant association between attack behaviours, escape attempts, and elevated corticosterone levels, by demonstrating that these behaviours can be used as indicators of decreased

**Funding:** This research was funded by Programma di sviluppo rurale 2014-2020. Misura 16. Innovazione e Cooperazione. Operazione 2.1 - Az. 2 - Progetti pilota – Piattaforma tecnologica bioeconomia. AlPiCoGriPi – Allevamento Pilota del Coniglio Grigio Piemontese: biodiversità, benessere e qualità della carne (Research agreement n. CUPJ69H22000000002). The funders had no role in study design, data collection and analysis, decision to publish, or preparation of the manuscript.

**Competing interests:** The authors have declared that no competing interests exist.

animals' well-being. Our findings underscore the importance of considering both the housing environment and the temporal dimension in the study of behaviour and welfare. This enables a comprehensive assessment of appropriate rearing management techniques. By understanding the social dynamics and stress sources within housing systems, farmers can implement measures to enhance animal welfare and create a conducive environment for the health and behaviour of rabbits.

## 1. Introduction

The European Union (EU) is the world's second-largest producer of meat rabbits, following China, and it is responsible for 93% of the global imports and exports in this industry [1]. Germany, Belgium, and Portugal are the primary importing countries, while Spain, Hungary, France, and Belgium are the major exporting countries [2]. Professional rabbit farming for commercial meat production is concentrated in Spain, France, and Italy, which together account for 83% of EU production. Specifically, Spain produces 48.5 million rabbits, France produces 29 million rabbits, and Italy produces 24.5 million rabbits [1]. In Italy, commercially reared rabbits are predominantly raised in standard wire cages. These housing conditions have been associated with elevated stress levels in the animals, which can compromise their overall welfare [3]. However, currently, there is no specific legislation at the EU level regarding rabbit housing, although some member states such as Italy, Germany, and Belgium have developed their own national legislation or recommendations. The Italian Ministry of Health has developed guidelines on welfare in rabbit breeding, which aim to standardize breeding practices and allow breeders to renew their cages in preparation for the adjustments required by the European Regulation. This Regulation will be established in accordance with the guidance provided by the EFSA (European Food Safety Authority) Scientific Opinion on the main critical points of cuniculture [4]. This report highlights the intensive single cage breeding system as a critical factor affecting the welfare of rabbits. The main concerns are related to narrow environments, high breeding densities, and the inability of animals to express social behaviours. In addition, public opinion, influenced by the perception of rabbits as pets, strongly advocates for the abandonment of single cages in rabbit breeding. As a result, there is a pressing need to identify and implement the most suitable alternative housing system to replace the traditional single cages in rabbit breeding.

Alternative rearing systems for rabbits encompass a variety of approaches designed to improve animal welfare and optimize production efficiency. Central to these systems are considerations such as cage/pen dimensions and environmental enrichment, which play pivotal roles in promoting the physical and psychological welfare of group-housed rabbits [5]. Cage or pen dimensions in alternative systems often prioritize spaciousness to allow for increased mobility and social interaction among rabbits [6]. Additionally, environmental enrichment strategies such as providing tunnels, platforms, or chew toys offer opportunities for mental stimulation, physical exercise, and natural behaviours such as burrowing and exploring. These elements are essential for ensuring the overall health of group-housed rabbits, contributing to their quality of life and productivity in alternative rearing systems [7]. In light of this, some tests might be helpful to evaluate the stress and fear response of naturally predated animals like rabbits [8]. Tonic immobility, observed in rabbits, serves as one such test. When faced with perceived threats or stressors, rabbits may enter a state of immobility, indicating their level of fear. This behaviour, characterized by sudden stillness and rigid posture, offers insights into

the mechanisms underlying fear and stress regulation in these prey animals [9]. However, alternative housing system solutions proposed at the European Community level and subsequently incorporated into the Ministerial Guidelines for rabbit breeding are not always beneficial for both animal welfare and production performance [10–13] and the advantages of alternative breeding systems for slow-growing local rabbit breeds remain uncertain. Additionally, aggressive social behaviours emerge with the onset of puberty, which occurs around 70 days of age [14]. These conditions contribute to chronic stress, which negatively impacts the immune system and the overall performance of the animals. The consequences include reduced growth, an increased incidence of injuries among rabbits, and elevated mortality rates [13]. Given their genetic peculiarities, the preservation of autochthonous slow-growing breeds is advantageous due to their high capacity to adapt and resist to the uprising concerns regarding climate change [15–17].

In the present study, we assessed the effects of three different housing systems (traditional single cage, group farming, and mixed system) on the welfare and behaviour of two Italian local rabbit populations the grey rabbit of Carmagnola (GC) and the grey rabbit of Monferrato (GM). They are characterized as a medium-slow growing breed with low reproductive efficiency. The primary objective of this study was to identify the most suitable housing systems for the two local rabbit populations. To achieve this, we evaluated their behaviours and welfare, short-term and long-term physiological stress indicators (i.e., salivary and hair corticosterone), blood stress indicators and we performed the tonic immobility test.

## 2. Materials and methods

### 2.1 Animals and housing

The research was conducted at the experimental farm of the Department of Veterinary Sciences, Turin University (Italy), from March to July 2022. All animals were handled in accordance with the recommendations of the Turin University Bioethics Committee (Protocol no. 0245520).

A total of 294 male weaned rabbits (35 days old) from two different grey rabbit populations, Carmagnola (GC, N = 147) and Monferrato (GM, N = 147), were randomly allocated to three breeding systems:

- Traditional single cage (Single): a total of 7 rabbits per breed were housed individually in cages measuring 500 x 250 x 300 mm, with a stocking density of 24 kg/m$^2$. Each cage was considered as an experimental unit (7 replicates).

- Group farming (Group): a total of 70 rabbits were housed in collective cages measuring 2 m$^2$ and a density of 15 kg/m$^2$ (7 replicates).

- Mixed pilot system (Mixed): a total of 70 rabbits were initially raised in groups with 7 rabbits per collective cage (15 kg/m$^2$). Once they reached sexual maturity (80 days old), they were transferred to single cages measuring 500 x 250 x 330 mm, with a stocking density of 24 kg/m$^2$ (7 replicates).

All the experimental groups were housed in the same artificially ventilated building with an airflow rate of 0.3 m/s. The environmental conditions, including temperature and relative humidity, were monitored, and controlled daily within the range of +15/+28˚C and 60% / 75%, respectively. The lighting schedule followed a 12-hour light/12-hour dark cycle (12L/12D). During the trial, from the time of weaning until commercial slaughtering age (100 days of age), the rabbits were provided with *ad libitum* access to feed and water. Daily health checks

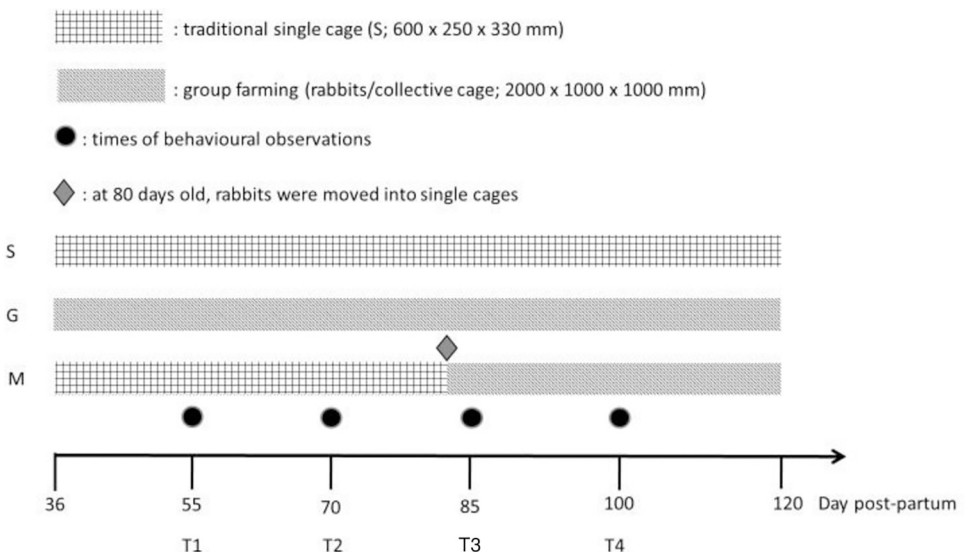

**Fig 1. Timing of experimental protocol of two autochthonous slow-growing grey rabbit populations housed in three different farming systems.** Data collection schedule of behavioural observation and TI test: T1: day 55, T2: day 70, T3: day 85 and T4: day 100.

were performed to monitor the health status of the rabbits and any deceased animals were removed.

## 2.2 Behavioural analysis

The behavioural patterns of the rabbits were recorded through direct observations conducted by two experienced operators who had undergone prior training together. The study used the Focal Animal Scan Sampling Method, as outlined by Lehner, which involves observing a designated individual (referred to as the focal animal) within a group and meticulously documenting its behaviour in real-time. This method enables a comprehensive understanding of individual behaviours, social dynamics, and responses to environmental stimuli within animal groups [18]. Behavioural observations were conducted at four different animals age: 55 (T1), 70 (T2), 85 (T3), and 100 (T4) days, as shown in Fig 1. The rabbits were observed during the daytime, between 9 a.m. and 11 a.m. and between 2 p.m. and 4 p.m. Therefore, their nocturnal activity was not recorded. The different times observed were chosen to be representative of a day and of a pre-sexual maturity situation and how this hormonal change impacted on animal overall welfare and behaviour.

   Prior to each observation period, the operators allowed a 5-minute adaptation period for the animals to acclimate to their presence. Data were recorded on a designated form. To determine the end of an observed behaviour, the operator waited for 10 seconds to see if the same behaviour was repeated. After the 10 seconds, any new behaviour observed was recorded. To develop the ethogram (Table 1), the following behaviours were recorded: kinetic activities (walking, running, jumping, turning on itself, and exploratory behaviours); feeding behaviours (eating and drinking); static activities (lying down, crouching, sitting, staying, and standing); comfort behaviours (self-scratching and self-grooming); stereotypical behaviours (smelling and biting bars); and social behaviours (attack, smelling others, allo-grooming, dominance features, and escape attempts). The specific ethogram was compiled based on Mugnai et al. [11] and further validated through preliminary observations. The behaviours were observed and registered as a frequency measure not considering duration. For each rabbit, the frequency of

**Table 1. Evaluated ethogram of two autochthonous slow-growing grey rabbit populations housed in three different farming systems.**

| Activities | Behaviour | Behaviour description |
|---|---|---|
| Kinetics | Walking | Any movement in any direction where two limbs are involved |
| | Running | Any movement in any direction where all four limbs are involved |
| | Jumping | Voluntary movements of jumping (almost 3) |
| | Turning on itself | A single movement to change the position from head to tail |
| | Exploratory | Walking and Sniffing |
| Feeding | Eating | Head above the feeder. Eating or chewing pellets |
| | Drinking | Drinking from water nipple |
| Statics | Laying down | Resting with chest or stomach on the floor. Fore limbs stretched in front of the body |
| | Crouching | Resting with chest or stomach on the floor. Hind and fore limbs crouched under body |
| | Sitting | Sat in upright position on hind limbs and fore limbs straight, but without bust touching the floor |
| | Staying | Standing still on four straight limbs |
| | Standing | Sitting in upright position on hind limbs and fore limbs straight |
| Comfort | Self-scratching | Licking, scratching, or nibbling of the body |
| | Self-grooming | |
| Stereotypies | Smelling bars | Smelling bars and cage floor insistently |
| | Biting bars | Licking or gnawing cage bars and scratching cage floor insistently |
| Social | Attack | Offensive moves, in which the doe attempts to bite its opponent |
| | Smelling others | Smelling another rabbit |
| | Allo-grooming | Licking, scratching, or nibbling another rabbit's body |
| | Dominance features | A rabbit that mounts, bites, or scratches another rabbit, or that sits with a tense body posture with erected ears and tail near to another doe |
| | Escape attempts | A rabbit that attempts to escape from another rabbit presence |

occurrence of each behaviour was calculated by dividing the number of times it was observed by the total number of observations. This frequency was then multiplied by 100 to obtain a percentage value.

## 2.3 Tonic immobility test

The Tonic Immobility (TI) test was conducted on the same observed animals for all four times (T1, T2, T3, and T4) (Fig 1). The rabbits were individually identified by a clipped area of fur on their right thigh. To perform the test, the operator gently removed the rabbit from its cage and induced immobility by turning the animal on its back while holding it in the operator's arms. The immobile rabbit was then placed on a plastic support surface, following the procedure described by Wilczyńska et al. [8]. A maximum of three attempts were made to induce immobility in each animal. The animals were not kept in the immobility condition for more than 5 minutes. The number of attempts required to induce immobility and the total duration of the condition were recorded for each animal. The assessment was carried out 48 hours subsequent to the behavioural evaluations by a trained operator, entirely unacquainted with the subjects, and exclusively dedicated to the execution of this experimental protocol.

## 2.4 Inter-observer reliability

Inter-observer reliability (IOR) is a crucial aspect for ensuring reliable behavioural and welfare assessments, as these assessments can be influenced by subjectivity and potential biases related

to the assessors' prior experience and level of empathy towards the animals [19]. To evaluate the reliability of the behavioural assessments conducted by the two observers involved in the study, the IOR was assessed. Three different methodologies were used to determine the extent to which these trained observers consistently observed and recorded data. Firstly, we conducted the Spearman correlation test to examine the correlation between the percentages of each observed behaviour. Secondly, we employed two agreement indexes: Bangdiwala's (B) [20] and Gwet's ($\gamma AC_1$) [21]. These indices were chosen based on Giammarino et al. [22], as they have demonstrated superior performance in evaluating animal welfare indicators. Both Bangdiwala's B index and Gwet's ($\gamma AC_1$) index measure the agreement proportion between the two observers, taking into account the total number of observations made by each observer. These indices range from 0 to 1, with 0 indicating disagreement, 0.5 representing neutrality, and values closer to 1 indicating higher agreement between the observers. Spearman correlation test was conducted using R software, Version 3.1.2 (R Core Team, 2014), with a significance level of $p \leq 0.05$ considered as statistically significant.

## 2.5 Corticosterone samples collection, hormonal evaluation, and blood stress indicators

To evaluate short-term and long-term physiological stress in rabbits, we assessed their saliva and hair corticosterone (CORT) levels, respectively. Samples were collected in the same rabbits that underwent the TI test at four times (T1, T2, T3, T4). Saliva was collected in the morning (at 9 a.m.) immediately after the TI test using a Salivette® with polyethylene pad (Sarsted AG & Co., Nümbrecht, Germany). A cut sampling swap was inserted into the corner of the rabbit's mouth with the use of a clamp and was subsequently chewed by the animal for 30–60 s. The amount of saliva taken from the individual rabbits varied within a range of 0.01–0.5 ml. The collected samples were immediately frozen and stored at -80˚C until analysis. Upon thawing, the stored samples were prepared for hormonal assays. To do this, the thawed samples underwent a 10-minutes centrifugation at 3500 rpm at 4˚C, and the saliva samples were then placed in Eppendorf® test tubes. The saliva samples were then placed in Eppendorf test tubes.

Hair samples were collected from the thigh, the area was first shaved as close as possible to the skin using previously cleaned scissors. The hair samples were then placed in labelled paper bags and stored under light protected and dry conditions at room temperature until the extraction. Hair CORT extraction was conducted following the method described by Meyer et al. [23] with modifications. 250 mg of hair were weighed and washed with 5mL of isopropanol. After 3 minutes of mixing, the excess solvent was removed, and the samples were dried under hood. The dried hair was cut into 1–3 mm-long fragments using scissors. Two 60 mg portions were then placed into a 5ml glass vial, and 3ml of methanol (Sigma Aldrich, IT) was added. Subsequently, the vials were incubated at 37˚C under an airstream suction hood for 18 hours and then centrifuged for 15 minutes at 2,500 rpm. The supernatant, collected in glass vials, was subjected to an airstream suction hood at 37˚C until it completely dried. These extracts were stored at -20˚Cte until analysis. Before the quantification of CORT, the extracted samples were reconstituted with 2mL of Assay Buffer (AB) (Arbor Assays™, Ann Arbor, MI, USA Saliva and hair CORT levels were determined with a multi-format commercial Elisa kit (K014; Arbor Assays™, Ann Arbor, MI, USA) validate for saliva, hair and other substrates. The inter- and intra-assay coefficients of variation were less than 10% for both saliva and hair. According to the manufacturer, the kit exhibited the following cross-reactivities: 100% with corticosterone, 18.9% with 1-dehydrocorticosterone, 12.3% with desoxycorticosterone, and 0.38% with cortisol. The results are reported as the amount of CORT in saliva (ng/mL) and in hair (ng/g).

Blood samples were collected from rabbits at 100 days of age to assess the heterophil/lymphocyte ratio (HLR) and oxidative stress parameters (i.e., UCARR and umol HClO/ml). H/L ratio Determination (CBC, complete blood count) was performed on EDTA blood samples with an automated laser analyser (ADVIA®120 Hematology System, Siemens Diagnostics). Automated differentials were validated by microscopic evaluation of blood smears stained with May Grünwald—Giemsa. One hundred leukocytes, including granular (heterophiles, eosinophils, and basophils) and non-granular (lymphocytes and monocytes) leukocytes, were counted on the slide and the H/L ratio was calculated.

## 2.6 Statistical analysis

The average percentage of each behaviour, the average seconds of the TI test, the saliva and hair concentration, and the blood stress indicators (expressed as mean ± standard deviation) were calculated for each rabbit populations (GC and GM), housing systems (Single, Group, Mixed), and times (T1, T2, T3, T4). Normality of data distribution was assessed using the Shapiro-Wilk test. We used the two-way analysis of variance (ANOVA) to evaluate the effects of the rabbit population, housing system, age, and their interactions. Multiple comparisons of the means were carried out by calculating the least significant difference with the Duncan test. Correlation analysis of the rabbits' behaviours was performed using Spearman's correlation coefficient rho and were corrected according to Bonferroni. Afterwards, the Generalized Linear Model (GLM) (gamma distribution with a log link function) was employed to explore the relationships between the rabbits' behaviours and CORT levels. Statistical analyses were conducted using R software, Version 3.1.2 (R Core Team, 2014), with a significance level of $p \leq 0.05$ considered as statistically significant.

# 3. Results

## 3.1 Inter-observer reliability

IOR of behavioural observers are presented in Table 2. The results demonstrated significant correlations between the percentages of all behaviours evaluated by the two observers. Additionally, the agreement indexes, Bangdiwala's B and Gwet's γAC1, provided further evidence of consistency in assessing these behaviours. The results consistently indicated values close to 1 for all behaviours, indicating a high level of agreement between the observers. The B index ranges from 0.798 (smelling bars) to 0.988 (attack and escape attempts), the γAC1 index ranges from 0.486 (smelling bars) to 0.977 (escape attempts).

## 3.2 Behavioural observations

The effects of population, housing system and age, and their interactions on percentage of behaviours of Carmagnola's (GC) and from Monferrato's (GM) grey rabbit are presented in Table 3. Overall, the housing system and the age of the rabbits had a greater effect on their behavioural patterns compared to the population to which they belong. The housing systems had an impact on the majority of behavioural patterns observed in rabbits (except for jumping, eating, sitting, staying, self-scratching, and escape attempts) and different effects of the housing system were observed depending on the specific behaviours examined. In particular, rabbits housed in Single exhibited higher frequencies of turning on itself ($p < 0.001$), laying down ($p < 0.001$), and drinking behaviours ($p = 0.045$). Conversely, rabbits in Group displayed increased kinetics activity, including running ($p = 0.0013$), walking ($p < 0.001$), and exploratory ($p = 0.009$) behaviours, as well as standing behaviour ($p = 0.03$) and social interactions such as attacks ($p < 0.001$) smelling others ($p < 0.001$), and dominance displays ($p < 0.001$).

**Table 2. Inter-observer reliability measures for two autochthonous slow-growing grey rabbit populations housed in three different farming systems.**

| Categories | Behaviours patterns | Spearman correlation | | B | $\gamma(AC_1)$ |
|---|---|---|---|---|---|
| | | rho | p-value | | |
| Kinetics | Walking | 0.20 | 0.005 | 0.943 | 0.932 |
| | Running | 0.78 | <0.001 | 0.890 | 0.773 |
| | Jumping | 0.27 | 0.008 | 0.943 | 0.932 |
| | Turning on itself | 0.68 | <0.001 | 0.821 | 0.715 |
| | Exploratory | 0.60 | <0.001 | 0.927 | 0.875 |
| Feeding | Eating | 0.93 | <0.001 | 0.971 | 0.853 |
| | Drinking | 0.97 | <0.001 | 0.985 | 0.841 |
| Statics | Laying down | 0.82 | <0.001 | 0.844 | 0.792 |
| | Crouching | 0.86 | <0.001 | 0.873 | 0.735 |
| | Sitting | 0.67 | <0.001 | 0.939 | 0.909 |
| | Staying | 0.65 | <0.001 | 0.869 | 0.773 |
| | Standing | 0.59 | <0.001 | 0.940 | 0.909 |
| Comfort | Self-scratching | 0.92 | <0.001 | 0.966 | 0.762 |
| | Self-grooming | 0.88 | <0.001 | 0.880 | 0.624 |
| Stereotypies | Smelling bars | 0.84 | <0.001 | 0.798 | 0.486 |
| | Biting bars | 0.76 | <0.001 | 0.893 | 0.796 |
| Social | Attack | 0.91 | <0.001 | 0.988 | 0.954 |
| | Smelling others | 0.83 | <0.001 | 0.942 | 0.796 |
| | Allo-grooming | 0.75 | <0.001 | 0.951 | 0.898 |
| | Dominance features | 0.90 | <0.001 | 0.987 | 0.932 |
| | Escape attempts | 0.71 | <0.001 | 0.988 | 0.977 |

Moreover, our analysis revealed that in Mixed, rabbits demonstrated more frequent crouching (p = 0.03), self-grooming (p = 0.004), and stereotypic activities such as smelling (p = 0.001) and biting bars (p < 0.001).

The age factor had an impact on the occurrence of certain behaviours, except for jumping, eating, drinking, laying down, crouching, staying, self-scratching, self-grooming, smelling bars, smelling others, and escape attempts. However, certain behaviours that showed variation over time were more prominently displayed during the early stages of the study (T1 and T2), including walking (p = 0.047), exploratory (p = 0.04), sitting (p < 0.001), standing (p < 0.001), self-grooming (p = 0.02), and allo-grooming (p < 0.001). On the other hand, social behaviours such as attack (p = 0.01) and dominance (p < 0.001) were more frequently observed during the final stages of the study (T3 and T4).

No differences in behavioural patterns were observed between the two populations, except for crouching (p = 0.04), self-scratching (p < 0.001), and self-grooming (p = 0.002), where the GC local population exhibited the highest percentage of these behaviours, except for crouching, which was more frequently observed in the GM population. Moreover, we observed a higher frequency if static behaviours in GM particularly in sitting (GM: 2.28±5.28; GC: 2.10 ±7.22), staying (GM: 3.50±7.16; GC: 2.90±7.75) and standing (GM: 1.39±4.37; GC: 1.05±3.73).

Although the population did not emerge as a significant factor for most behaviours, when examined as single factor, our findings revealed numerous effects of population, both in interaction with the system and with age. Our analysis of exploratory (p = 0.005), laying down (p = 0.01), crouching (p = 0.02), staying (p = 0.035), biting bars (p = 0.03), and attack (p = 0.04) behaviours revealed an interaction between population and housing system. These findings indicate that the combination of population and specific housing system had distinct

**Table 3. Effect of population, housing system, age, and their interactions on behaviours (%) of two autochthonous slow-growing grey rabbit populations housed in three different farming systems.**

| Behaviour | Population (Po) | | Housing System (Sy) | | | Age (A) | | | | p-value | | | | | |
|---|---|---|---|---|---|---|---|---|---|---|---|---|---|---|---|
| | GC | GM | Single | Group | Mixed | T1 | T2 | T3 | T4 | Po | Sy | A | Po X Sy | Po X A | Sy X A |
| Running | 0.23 ±1.37 | 0.42 ±2.24 | 0[b] | 0.98[a] ±3.13 | 0[b] | 0[a] | 0[a] | 0.86[b] ±2.67 | 0.27[a] ±2.15 | 0.460 | 0.001 | 0.047 | 0.06 | 0.007 | 0.830 |
| Walking | 2.76 ±5.90 | 4.28 ±10.1 | 0.95[b] ±7.96 | 7.72[a] ±9.65 | 1.89[b] ±5.04 | 8.72[a] ±14.7 | 1.41[b] ±3.77 | 2.62[b] ±5.62 | 3.92[b] ±8.56 | 0.186 | <0.001 | <0.001 | 0.068 | 0.007 | 0.220 |
| Jumping | 0.29 ±1.49 | 0.37 ±2.27 | 0.20 ±1.70 | 0.55 ±2.04 | 0.23 ±1.99 | 0 | 0.27 ±2.15 | 0.51 ±1.95 | 0.37 ±2.11 | 0.753 | 0.480 | 0.680 | 0.160 | 0.090 | 0.060 |
| Turning on itself | 4.33 ±7.78 | 3.38 ±7.77 | 7.33[a] ±10.4 | 1.37[b] ±3.67 | 2.86[b] ±6.44 | 0.19[c] ±1.07 | 4.51[ab] ±8.07 | 7.37[a] ±10.0 | 1.52[bc] ±4.72 | 0.373 | <0.001 | <0.001 | 0.086 | 0.032 | 0.520 |
| Exploratory | 1.12 ±3.63 | 0.78 ±2.83 | 0[b] | 1.54[a] ±3.74 | 1.31[a] ±4.09 | 1.14[ab] ±3.58 | 1.90[a] ±4.56 | 0.37[b] ±2.06 | 0.49[ab] ±2.17 | 0.468 | 0.009 | 0.040 | 0.005 | <0.001 | 0.002 |
| Eating | 5.31 ±14.89 | 2.47 ±7.48 | 4.51 ±11.5 | 2.86 ±7.08 | 4.31 ±15.5 | 2.15 ±5.85 | 4.67 ±12.1 | 2.63 ±5.87 | 5.26 ±17.0 | 0.084 | 0.667 | 0.493 | 0.288 | 0.293 | <0.001 |
| Drinking | 2.77 ±7.44 | 3.00 ±8.84 | 4.12[a] ±11.4 | 3.61[a] ±7.37 | 0.93[b] ±3.25 | 1.63 ±7.43 | 4.17 ±11.62 | 3.05 ±5.19 | 2.06 ±6.57 | 0.837 | 0.045 | 0.428 | 0.121 | 0.506 | <0.001 |
| Laying down | 15.2 ±31.9 | 18.2 ±35.1 | 30.8[a] ±41.3 | 14.8[b] ±33.2 | 4.65[b] ±15.2 | 4.44 ±19.0 | 20.22 ±37.0 | 15.9 ±31.9 | 20.3 ±36.1 | 0.514 | <0.001 | 0.143 | 0.010 | 0.090 | 0.010 |
| Crouching | 15.3[a] ±27.7 | 24.5[b] ±37.1 | 13.17[b] ±25.9 | 18.9[ab] ±32.5 | 27.7[a] ±38.2 | 19.6 ±22.5 | 20.7 ±37.2 | 16.8 ±30.6 | 22.4 ±35.7 | 0.045 | 0.030 | 0.827 | 0.020 | 0.270 | 0.005 |
| Sitting | 2.28 ±5.28 | 2.10 ±7.22 | 2.29 ±5.86 | 1.84 ±5.87 | 2.44 ±7.18 | 9.20[a] ±12.2 | 1.59[b] ±4.39 | 0.83[b] ±3.20 | 0.64[b] ±2.93 | 0.832 | 0.840 | <0.001 | 0.370 | 0.042 | 0.850 |
| Staying | 3.50 ±7.16 | 2.90 ±7.75 | 3.07 ±7.84 | 2.60 ±7.18 | 3.92 ±7.37 | 5.21 ±9.01 | 3.57 ±7.75 | 1.77 ±5.43 | 3.24 ±7.94 | 0.567 | 0.570 | 0.210 | 0.035 | 0.011 | 0.090 |
| Standing | 1.39 ±4.37 | 1.05 ±3.73 | 0.35[b] ±2.98 | 2.13[a] ±4.92 | 1.18[ab] ±3.89 | 4.36[a] ±8.02 | 1.15[b] ±3.59 | 0.64[b] ±2.19 | 0.30[b] ±1.66 | 0.536 | 0.030 | <0.001 | 0.110 | 0.110 | 0.800 |
| Self-scratching | 5.95[a] ±9.89 | 2.16[b] ±5.64 | 4.52 ±8.44 | 4.92 ±7.99 | 2.71 ±8.27 | 1.84 ±4.44 | 4.16 ±8.76 | 5.68 ±9.52 | 3.42 ±7.65 | <0.001 | 0.246 | 0.183 | 0.484 | 0.771 | 0.812 |
| Self-grooming | 13.1[a] ±13.3 | 7.59[b] ±12.9 | 11.5[ab] ±14.0 | 6.28[b] ±10.3 | 13.3[a] ±14.6 | 13.08[ab] ±15.0 | 13.21[a] ±14.9 | 10.14[ab] ±11.2 | 6.40[b] ±12.2 | 0.002 | 0.004 | 0.028 | 0.394 | 0.595 | 0.031 |
| Smelling bars | 13.7 ±14.7 | 12.0 ±14.7 | 13.5[a] ±14.7 | 8.09[b] ±10.8 | 17.1[a] ±16.7 | 13.9 ±12.8 | 11.8 ±14.8 | 14.3 ±13.7 | 12.0 ±16.5 | 0.408 | 0.001 | 0.759 | 0.892 | 0.757 | <0.001 |
| Biting bars | 3.16 ±8.25 | 4.60 ±11.4 | 1.86[b] ±5.39 | 1.95[b] ±5.68 | 7.84[a] ±14.7 | 2.28[ab] ±6.27 | 0.38[b] ±2.16 | 5.77[a] ±9.71 | 6.29[a] ±14.5 | 0.296 | <0.001 | 0.002 | 0.033 | 0.859 | <0.001 |
| Attack | 0.47 ±3.43 | 1.66 ±6.36 | 0[b] | 3.21[a] ±8.53 | 0[b] | 0[a] | 0[a] | 0.91[b] ±4.31 | 2.83[a] ±8.36 | 0.093 | <0.001 | 0.010 | 0.046 | 0.515 | <0.001 |
| Smelling others | 3.12 ±8.71 | 3.28 ±6.91 | 0.17[b] ±1.49 | 7.24[a] ±11.0 | 2.18[b] ±6.02 | 1.84 ±4.44 | 3.57 ±4.44 | 2.45 ±6.61 | 4.25 ±10.7 | 0.886 | <0.001 | 0.448 | 0.244 | 0.188 | <0.001 |
| Allo-grooming | 3.07 ±7.62 | 1.71 ±6.62 | 0.17[b] ±1.49 | 1.67[b] ±4.92 | 5.33[a] ±10.7 | 10.3[a] ±13.5 | 2.60[b] ±6.80 | 0.50[b] ±1.98 | 0.11[b] ±0.86 | 0.162 | <0.001 | <0.001 | 0.348 | 0.184 | <0.001 |
| Dominance | 0.98 ±4.13 | 0.63 ±3.06 | 0[b] | 2.42[a] ±6.00 | 0[b] | 0[b] | 0[b] | 2.54[a] ±6.32 | 0.28[b] ±1.57 | 0.473 | <0.001 | <0.001 | 0.572 | 0.937 | <0.001 |
| Escape attempts | 0.00 | 0.49 ±2.97 | 0 | 0.73 ±3.63 | 0 | 0 | 0 | 0.18 ±1.43 | 0.67 ±3.67 | 0.090 | 0.061 | 0.293 | 0.063 | 0.280 | 0.261 |
| | | | | | Salivary and hair corticosterone | | | | | | | | | | |
| CORT saliva (ng/mL) (ng/g) | 15.0 ±5.45 | 17.67 ±8.84 | 16.14[a] ±7.36 | 18.95[ab] ±10.80 | 13.99[b] ±4.80 | 9.67[c] ±5.03 | 15.01[b] ±3.60 | 21.14[a] ±9.78 | 19.53[ab] ±7.88 | 0.080 | 0.020 | <0.001 | 0.300 | 0.730 | 0.002 |
| CORT hair (ng/g) | 13.10 ±4.84 | 14.47 ±5.70 | 12.06[b] ±5.04 | 14.87[a] ±5.68 | 14.41[ab] ±4.87 | 13.68[ab] ±6.15 | 15.69[a] ±5.27 | 11.56[b] ±3.27 | 14.20[ab] ±5.67 | 0.160 | 0.030 | 0.020 | 0.460 | 0.400 | 0.030 |

GC = Carmagnola grey rabbit; GM = Monferrato grey rabbit. T1 = 55 days; T2 = 70 days; T3 = 85 days; T4 = 100 days. Means with superscript letters (a, b, c) denote significant differences (p < 0.05).

effects on these behaviours. Similarly, we found interactions between the population and the age in running (p = 0.007), walking (p = 0.007), turning on itself (p = 0.032), exploratory (p < 0.001), sitting (p = 0.042), staying (p = 0.011), biting bars (p = 0.03) and attack (p = 0.04) behaviours, indicating that the effects of population and age combined had a notable impact on these behaviours. Finally, the analysis of exploratory (p = 0.002), eating (p < 0.001), drinking (p < 0.001), laying down (p = 0.01), crouching (p = 0.005), self-grooming (p = 0.03), smelling bars (p < 0.001), biting bars (p < 0.001), attack (p < 0.001), smelling others (p < 0.001), allo-grooming (p < 0.001), and dominance (p < 0.001) behaviours revealed significant interactions between housing system and age. Furthermore, it is important to consider the timing of observations. In the case of the Mixed, it should be noted that at T3 and T4, rabbits were

individually housed in single cages to mitigate potential negative effects like conspecific aggressivity and agonistic behaviours typically associated with sexual maturity [24]. Although the observers did not directly observe rabbits plucking their fur, the presence of fur found under the collective cage of CG and CM rabbits at T2 suggested the occurrence of this behaviour.

### 3.3 Salivary and hair corticosterone and blood stress indicators

The effects of populations, housing systems, and age on salivary and hair CORT levels are presented in Table 3. The populations did not show significant effects on both salivary and hair CORT levels. However, housing systems and the age of the animals, along with their interactions, had a significant impact on both short-term and long-term physiological stress in rabbits (Fig 2). Rabbits hosted in Group exhibited higher levels of both salivary and hair CORT. Higher levels of salivary CORT were observed in Single compared to Mixed system. Conversely, in the Mixed system, rabbits displayed higher levels of hair CORT compared to the Single system. Regarding the age factor, salivary CORT levels increased at the end of the study (T3 and T4), indicating elevated short-term stress levels during those periods (Fig 2). On the other hand, hair CORT showed higher levels at T2 and T4, suggesting a different pattern of long-term physiological stress (Fig 2).

Generalized linear models (GLM) were used to examine how the saliva and hair CORT levels varied in relation to the rabbits' behaviours. To avoid potential multicollinearity issues, behaviours that were found highly and significantly correlated were excluded from the models

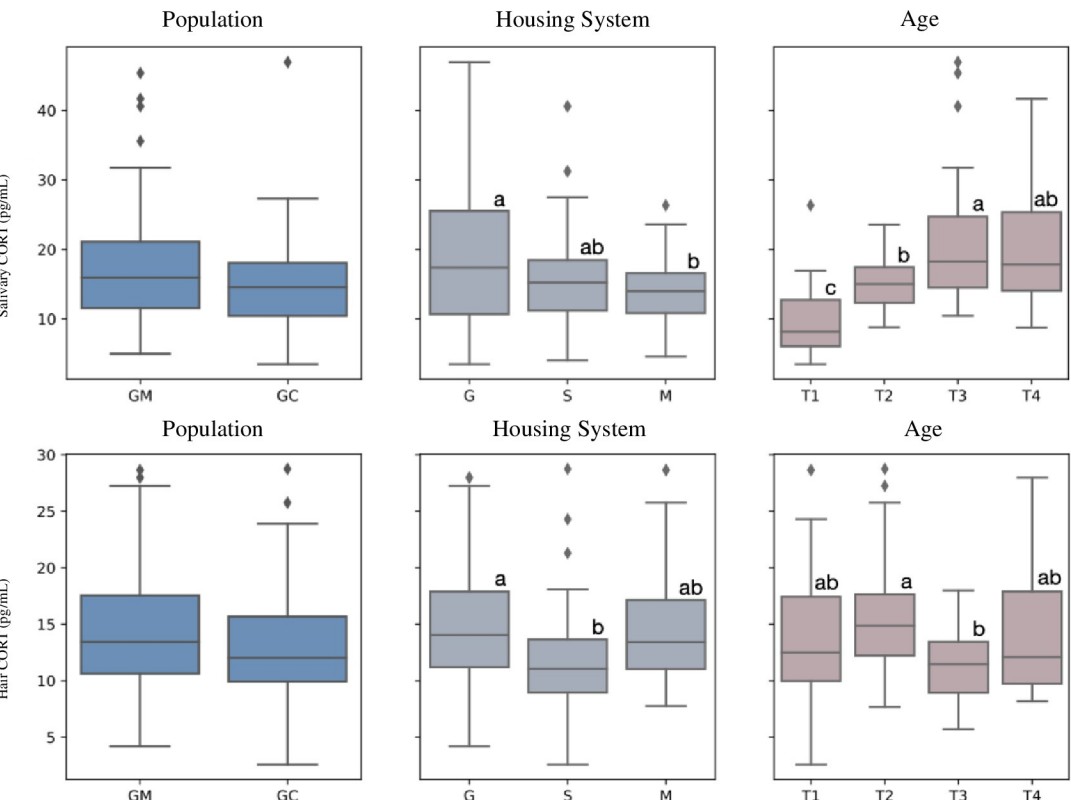

**Fig 2. Salivary and hair CORT levels of two autochthonous slow-growing grey rabbit populations housed in three different farming systems on population, housing system, and age.** The box and whisker plots illustrate the interquartile range, and the black lines indicate the median. The error bars extend from the box to the highest and lowest values. The diamonds indicate the outlier's data.

**Table 4. Generalized Linear Model for salivary corticosterone and hair corticosterone levels.** The dependent variable is the salivary CORT, and the independent variables (predictors) are the rabbit behaviours.

| Variable | Salivary corticosterone levels | | | | Hair corticosterone levels | | | |
|---|---|---|---|---|---|---|---|---|
| | Estimate | Standard error | z-value | p-value | Estimate | Standard error | z-value | p-value |
| Intercept | 19602 | 1494 | 13121 | <0.001 | 14175 | 1025 | 13830 | <0.001 |
| Exploratory | -0.147 | 0.240 | -0.611 | 0.541 | 0.106 | 0.165 | 0.646 | 0.518 |
| Eating | -0.058 | 0.065 | -0.894 | 0.371 | -0.049 | 0.045 | -1104 | 0.269 |
| Crouching | -0.042 | 0.026 | -1658 | 0.097 | -0.001 | 0.018 | -0.044 | 0.965 |
| Sitting | -0.225 | 0.095 | -2358 | 0.018 | -0.032 | 0.065 | -0.495 | 0.621 |
| Staying | 0.021 | 0.107 | 0.198 | 0.843 | 0.003 | 0.073 | 0.044 | 0.965 |
| Self-Grooming | -0.140 | 0.057 | -2443 | 0.015 | -0.013 | 0.039 | -0.342 | 0.732 |
| Biting-Bars | -0.139 | 0.078 | -1793 | 0.073 | -0.068 | 0.053 | -1292 | 0.196 |
| Attack | 0.413 | 0.156 | 2641 | 0.008 | 0.227 | 0.107 | 2122 | 0.034 |
| Dominance | 0.214 | 0.224 | 0.959 | 0.338 | -0.202 | 0.154 | -1321 | 0.186 |
| Escape-Attempts | 0.296 | 0.399 | 0.742 | 0.458 | 0.559 | 0.274 | 2041 | 0.041 |

(S1 Fig in S1 File). The results revealed a significant increase in both salivary and hair CORT levels as attack behaviours increased. Conversely, salivary CORT levels increased when both sitting and self-grooming behaviours decreased (Table 4). Additionally, hair CORT showed an increase in escape attempts (Table 4).

Overall, no significant differences were found in the blood stress indicators of rabbits when investigating the effects of population, housing system, and age of rabbits (S2 Table in S1 File).

### 3.4 Tonic immobility

The analysis investigating the effects of population, housing system, and age on tonic immobility in rabbits demonstrated that there were no statistically significant differences in TI for each attempt. All the data were combined and pooled together to calculate the mean value, which also showed no significant differences. Additionally, the interactions between population and housing system, population and age, and housing system and age were not found to be significant (S1 Table and S2 Fig in S1 File). Overall, no significant differences were found in the blood stress indicators of rabbits when investigating the effects of population, housing system, and age of rabbits (S2 Table in S1 File).

### 4. Discussion

It is well known that animal behaviour is influenced by the environment, and understanding how the environment affects animals is crucial for assessing and quantifying welfare. In this study, we investigated the impact of three different housing systems (single cage, group farming, mixed system) on the welfare and behaviour of two Italian local rabbit populations, taking into account the age of the rabbits as well. We achieved a high level of inter-observer reliability in behavior assessment, which revealed significant agreement among the methods employed. This reinforces the validity and reliability of our findings and the accuracy of the behavioral data.

Regarding differences in the behaviours exhibited by the two rabbit populations (i.e., GM and GC), we observed that only specific behavioural patterns, particularly crouching behaviour and comfort activities, differed. We observed a significant population effect on comfort activities, with GC rabbits displaying a higher propensity for self-scratching and self-grooming compared to GM rabbits. These results suggest the possible presence of genetic differences between the two populations that contribute to their distinct behavioural patterns. We hypothesize that

these differences may derive from selective breeding practices or environmental factors specific to each population. Further investigation into the genetic factors contributing to these behavioural differences could provide valuable insights into the underlying mechanisms shaping rabbit behaviour.

Beyond population differences and possible genetic variations, environmental enrichment plays a key role in promoting natural behaviours, providing animals with a greater number of behavioural opportunities [25]. Rabbits, like other animals, have specific behavioural expectations in relation to their surroundings, and the environmental conditions in which they are reared have a direct impact on their behaviour. Accordingly, we found significant effects of housing system on the behavioural patterns of both populations of grey rabbits. Rabbits in Group exhibited a broader range of behaviours, with a higher percentage of kinetic activities like running, walking, and exploratory behaviours. This finding aligns with previous research by Dal Bosco et al. [26], L. Lambertini and Formigoni [27], Princz et al. [28]; Trocino et al. [29], who also reported increased movement in rabbits housed in group systems, and a negative correlation between movement and eating activity. These findings indicate that group housing offers a more stimulating and dynamic environment for rabbits, leading to a broader range of physical activities. This was associated with a reduction in stereotypical behaviours, decreased time spent on feeding and resting, and an increase in social activities, exploration, and aggressiveness, in line with previous research [26, 28]. On the other hand, rabbits housed in Single exhibited higher frequencies of turning on itself, laying down, and drinking behaviours, while rabbits in Mixed displayed more crouching, self-grooming, and stereotypic activities such as smelling and biting bars. These observations suggest that the Mixed system may not provide an optimal environment for rabbits, given the increased occurrence of stereotypical behaviours. In the case of rabbits housed in Single, their behavioural repertoire is limited due to the spatial constraints of their environment. Social activities are restricted, as rabbits have limited opportunities for performing behaviours such as smelling others and allo-grooming, especially when neighbouring rabbits are housed in adjacent cages [30]. Research indicates that anxiety symptoms are often linked to restrictive repetitive behaviours (RRBs), particularly when animals engage in repetitive behaviours consistently [26, 27]. This may explain why rabbits in Single-cages exhibit anxious repetitive behaviours, such as bar biting and sniffing, with the latter two classified as stereotyped behaviours [3].

The impact of age on behaviours, particularly kinetic activities such as walking and exploratory behaviours, was significant, in addition to the effect of the housing system. Rabbits housed in Single exhibited limited opportunities for kinetic activities, except for turning on themselves, as previously discussed. Conversely, rabbits in collective cages, including both in Group and Mixed systems, had more available space for movement, particularly at younger ages (T1 and T2). The contrast in available space for kinetic activities among the three housing types underscores the importance of housing design in facilitating rabbit behaviour. Single housing, while offering individual space, may restrict movement due to spatial limitations, leading to predominantly stereotypic behaviours such as turning. This difference in space availability highlights the potential benefits of Group and Mixed housing configurations, as they better accommodate the locomotor needs of rabbits, particularly during early developmental stages. Moreover, it is well-known that locomotor activity in rabbits tends to decrease with age. Consistent with the findings of Trocino et al., [29] our study revealed that the occurrence of running behaviour was influenced by both the housing system and age, with higher frequencies observed in older rabbits (T3 and T4) within the Group system. The tonic immobility test did not show any significant differences among the housing systems, age of the rabbits, and populations. This lack of significant differences could be attributed to the regular handling and interaction of rabbits by the farmer during routine management practices. It is

widely known that animals can gradually become habituated to human presence and contact, resulting in a decrease in fear responses over time. One limitation of our study is that our sampling method does not fully account for the circadian variations in corticosterone levels. Corticosterone levels naturally fluctuate throughout the day, and our fixed sampling time may not capture these dynamic changes accurately. As a result, while our morning saliva samples reflect the stress levels from the preceding night, they may not provide a comprehensive view of the fluctuations in corticosterone that occur over a 24-hour period. All observed animals were males, thus there was no sex variability in our study. However, sex differences could have a significant impact on behaviour and may be an area of interest for future studies. Furthermore, the total number of animals observed was 294. While this number may seem substantial, increasing the sample size in future studies could enhance the representativeness of our results and provide a better understanding of the observed behavioural patterns.

Our results on corticosterone levels provide valuable insights into the physiological stress experienced by the rabbits in different housing systems, which is consistent with previous studies [3]. Prior studies investigating the diurnal rhythm of salivary corticosterone concentration in rabbits have highlighted fluctuations in their stress hormone levels over the course of a day. Notably, research indicates highest corticosterone levels between 12:00 and 15:00 [31]. These findings underscore the dynamic nature of stress regulation in rabbits and provide valuable insights into the temporal patterns of their physiological responses to environmental stimuli. Such understanding contributes significantly to our comprehension of the adaptive mechanisms employed by rabbits in coping with varying stressors encountered in their natural habitat. Rabbits housed in Group system exhibited higher levels of both salivary and hair CORT, indicating an increased stress response in this housing condition. This might be attributed to factors such as social dynamics, competition for resources, or other stressors associated with group housing. This finding is consistent with previous studies that have reported increased stress levels in group-housed animals [32] including rabbits [10, 33] due to factors such as social hierarchy and environmental challenges. The Single system was associated with higher levels of salivary CORT compared to the Mixed system. This result suggests that individual housing might lead to acute stress responses in rabbits, possibly due to the limited opportunities for social interactions and environmental enrichment in single cages [34]. The lower hair CORT levels (i.e., lower chronic stress) observed in rabbits housed in Single could be attributed to a potential coping response, as suggested by Mugnai et al. [3]. The coping response refers to behaviours that appear to attenuate stressor-induced physiological responses [35] by exerting a calming effect [36]. Rabbits in single cages exhibited a higher frequency of stereotypical behaviours, notably turning on itself. It is plausible that this behaviour triggered a calming effect, contributing to the maintenance of lower CORT levels in these animals. We acknowledge that a limitation of using corticosterone in hair as a biomarker for chronic stress is the individual differences in hair coloration. Even within a population of rabbits with uniform hair color, there may be individual variations that require further investigation. Regarding the age factor, we observed that salivary CORT levels increased at the end of the study (T3 and T4), indicating elevated short-term stress levels during those periods. This observation could be attributed to factors such as the attainment of sexual maturity, which may have triggered acute stress responses in the rabbits. On the other hand, hair corticosterone levels showed higher values at T2 and T4, suggesting a different pattern of long-term physiological stress. This pattern could be influenced by the cumulative effects of chronic stressors experienced by the rabbits over time, which might result in a delayed impact on hair CORT levels. Moreover, the influence of age and its interaction with the housing system had a significant effect on allo-grooming behaviour. As the rabbits aged, the occurrence of this cohesive social behaviour decreased, particularly at T3 and T4 when the rabbits reached sexual maturity.

Instead, aggressive behaviours such as attack, dominance features, and escape attempts became more prevalent. These findings align with previous studies conducted by Lambertini et al. [27], Dalle Zotte and Szendro [37], and Trocino et al. [38], which reported an increased risk of aggression among rabbits as they approached sexual.

To investigate the potential connections between specific behaviours and physiological stress responses in rabbits we used a Generalized Linear Models (GLM) in which the significant increase in both salivary and hair CORT levels as aggressive behaviours increased suggests a potential link between aggressive interactions and both acute and chronic stress reactivity in rabbits. This finding aligns with previous research in other animal species, indicating that aggressive behaviours can elicit physiological stress responses [39, 40]. On the other hand, we found that the decrease in self-grooming and sitting behaviours was associated with an increase in salivary CORT levels. These behaviours are often associated with relaxation and comfort, and their decrease may indicate higher acute stress levels in the rabbits. Additionally, the GLM revealing an increase in hair CORT levels in response to escape attempts highlights the potential long-term effects of stress on the rabbits' physiology. Escape attempts are indicative of aversive or challenging situations, and the observed association with hair CORT levels may suggest that these stressful experiences have a lasting impact on the animals' stress hormone levels. By considering both behavioural and physiological indicators, we can better assess the welfare and well-being of rabbits in various environments and identify areas where improvements can be made to enhance their living conditions.

## 5. Conclusions

Our research emphasizes the importance of observing both the behaviour and physiological stress markers of rabbits over time to understand their well-being in different housing systems. We have highlighted that the type of housing significantly affects various behaviours in rabbits. For instance, group farming fosters social bonding but can also lead to increased levels of chronic and acute stress in rabbits. Conversely, rabbits in solitary cages may experience acute stress due to loneliness and confinement. These differences arise from both social and physiological changes in rabbits, which should be consider when selecting the appropriate housing system. However, it's essential to acknowledge some limitations in our study, analyzing rabbit behaviour during night-time, considering their nocturnal nature, could offer a more complete picture of their behavioural patterns and stress responses. Furthermore, the timing of observations plays a crucial role in understanding how housing systems influence behaviour. Our statistical analyses provide deep insights into the complex relationship between behaviour and stress physiology in rabbits, uncovering underlying stressors and adaptive coping mechanisms across different farming conditions. The relationship we've identified between aggressive behaviours, escape tendencies, and cortisol levels present promising avenues for identifying key behavioural indicators. Armed with a deeper understanding of social dynamics and stress factors within farming systems, our findings equip farmers with targeted interventions to enhance animal welfare and create an environment conducive to optimal health and behaviour.

## Supporting information

**S1 File.**
(DOCX)

## Author Contributions

**Conceptualization:** Cecilia Mugnai.

**Data curation:** Laura Ozella, Riccardo Crosetto, Edoardo Fiorilla.

**Formal analysis:** Laura Ozella, Edoardo Fiorilla.

**Funding acquisition:** Patrizia Ponzio, Cecilia Mugnai.

**Investigation:** Stefano Sartore, Elisabetta Macchi, Isabella Manenti, Silvia Mioletti, Barbara Miniscalco, Riccardo Crosetto, Cecilia Mugnai.

**Methodology:** Laura Ozella, Stefano Sartore, Edoardo Fiorilla, Cecilia Mugnai.

**Project administration:** Cecilia Mugnai.

**Supervision:** Stefano Sartore, Cecilia Mugnai.

**Validation:** Laura Ozella, Cecilia Mugnai.

**Writing – original draft:** Laura Ozella, Edoardo Fiorilla, Cecilia Mugnai.

**Writing – review & editing:** Laura Ozella, Stefano Sartore, Elisabetta Macchi, Isabella Manenti, Silvia Mioletti, Barbara Miniscalco, Edoardo Fiorilla, Cecilia Mugnai.

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
