## [Decision Letter · Decision Letter 0]

12 Mar 2024

PONE-D-24-03037Behaviour and Welfare Assessment of Autochthonous Slow-Growing Rabbits: The Role of Housing SystemsPLOS ONE

Dear Dr. Fiorilla,

Thank you for submitting your manuscript to PLOS ONE. After careful consideration, we feel that it has merit but does not fully meet PLOS ONE’s publication criteria as it currently stands. Therefore, we invite you to submit a revised version of the manuscript that addresses the points raised during the review process.

Please carefully address the general and specific comments of the reviewers.

We look forward to receiving your revised manuscript.

Kind regards,

Harvie P. Portugaliza, D.V.M., Ph.D.

Academic Editor

PLOS ONE

Journal Requirements:

"This research was funded by Programma di sviluppo rurale 2014-2020. Misura 16. Innovazione e Cooperazione. Operazione 2.1 - Az. 2 - Progetti pilota – Piattaforma tecnologica bioeconomia. AlPiCoGriPi – Allevamento Pilota del Coniglio Grigio Piemontese: biodiversità, benessere e qualità della carne (Research agreement n. CUPJ69H22000000002)."

4. In the online submission form, you indicated that [Data will be made available on request]. 

6. Please upload a copy of Figure 3, to which you refer in your text on page 9. If the figure is no longer to be included as part of the submission please remove all reference to it within the text.

Additional Editor Comments:

Please carefully address the general and specific comments of the reviewers.

Reviewers' comments:

Reviewer's Responses to Questions

**Comments to the Author**

1. Is the manuscript technically sound, and do the data support the conclusions?

Reviewer #1: Partly

Reviewer #2: Partly

2. Has the statistical analysis been performed appropriately and rigorously? 

Reviewer #1: Yes

Reviewer #2: Yes

3. Have the authors made all data underlying the findings in their manuscript fully available?

Reviewer #1: Yes

Reviewer #2: Yes

4. Is the manuscript presented in an intelligible fashion and written in standard English?

Reviewer #1: Yes

Reviewer #2: Yes

5. Review Comments to the Author

Reviewer #1: GENERAL COMMENT

The present work was aimed at evaluating the behaviour and some physiological indicators of stress in two Italian local breeds of rabbits reared in conventional and alternative housing systems.

The topic of the present work is interesting, but the manuscript needs a revision to clarify some key points.

In my opinion the work would have been more impactful with the presence of a commercial hybrids used as control. This comparative approach would allow for a better understanding of how different genotypes respond to challenging conditions and could contribute to discussions on the sustainability and resilience of various livestock farming systems. It may be beneficial for future research to consider such comparisons to enhance the overall impact and relevance of the findings.

The study could benefit from additional recordings to offer a more comprehensive evaluation of the impact of housing systems on animal welfare. Specifically, the absence of lesion scoring is a notable limitation, as it hinders a thorough assessment of potential physical impacts on the animals. Furthermore, including basic data on live weight, growth rate, and feed efficiency would enhance the overall understanding of the animals' physiological responses and performance under different housing conditions. Incorporating these additional measures would contribute valuable insights to the study, providing a more holistic perspective on the welfare and productivity implications of diverse housing systems.

One notable limitation of this paper pertains to certain gaps in the materials and methods section. Specifically, there is a need for clarification regarding key information, such as the number of replicates per group, the total number of observed rabbits, and the sex of the animals. These details are crucial for a comprehensive understanding of the experimental design and the robustness of the study’s findings. Addressing these omissions would enhance the transparency and reliability of the research, providing readers and fellow researchers with essential insights into the experimental setup and the transferability of the results.

SPECIFIC COMMENTS

ABSTRACT

L13: from this line the authors introduce the term “populations”. It should be clarified if these are two populations of the same breed or if they are two different breeds.

L13: The authors could introduce the acronym for the two populations.

L16: before the results it would be useful to report some other information regarding materials and methods, i.e., number of animals and replicates per group, duration of the trial, age of the animals.

L25: did the rabbits reach the sexual maturity during the trial?

L26-28: is this the first study that demonstrate the correlation between agonistic behaviours and corticosterone levels?

INTRODUCTION

L41-43: this sentence does not seem particularly useful. It could be cancelled.

L54-55: there is recent scientific literature regarding the characteristics of alternative rearing systems including information on cage/pen dimensions and environmental enrichment for group-housed rabbits. It could be useful to include a brief description and cite pertinent articles.

L63-65: it is clear, but why did you choose two populations of the same breed? It should be motivated in the introduction because in terms of impact it would be more useful to compare two different local breeds or a local breed with a commercia genotype. This is what I would have expected reading this sentence.

L67: it could be useful to introduce the name of the two populations and their acronyms here.

MATERIALS AND METHODS

L77-86: It is necessary to provide clear information regarding the number of replicates per group (cages/pens), and the sex of the animals.

L79-83: For the housing systems I would use acronyms that can be easily remembered and interpreted, e.g., Single, Group, Mixed.

L91: What exactly was the slaughter age?

L91-93: Since the authors monitored the health (I think morbidity and mortality) I would expect a description of these results in the next section (Results).

L96-97: Please specify the exact number of rabbits observed per group.

L98-100: This is a weakness of the work since it is widely known that rabbits are active during night. The authors should consider this aspect when discussing the results.

L175-184: Did you perform a check of data distribution? Which distribution was set for the model?

RESULTS

L195: The acronyms should appear in the abstract and then the first time you define the population in the main text (L67).

L199-234: Reporting p-values and the term “significant” in redundant. It would be more useful to report some relative numerical variations among groups.

L236-260: For a more detailed and informative presentation of the results, providing relative numerical variations among groups when significant would enhance the clarity and precision of the findings. This approach can help readers better understand the magnitude and direction of the observed effects within and across different groups, contributing to a more comprehensive interpretation of the study outcomes.

DISCUSSION

The structure of the discussion should be changed. It would be better to discuss the results according to the main effects analysed (population, housing systems and age).

Probably, the most interesting interaction is that between housing system and age due to the presence of the Mixed system. The discussion regarding this interaction should be expanded.

Three other key aspects should be included and considered in the discussion: 1) the sex of the animals; 2) the number of animals observed (if I understood correctly, it was not that high); 3) the period of observation (limited to direct observations during daylight).

L266-269: there are more pertinent papers that report the effect of the housing system on rabbits’ behaviour and welfare.

L300-303: these results are not reported in the previous section. Please implement the section Results before their discussion. Furthermore, there is no information regarding morbidity and mortality that are crucial for an evaluation of the animal welfare state.

L305-306: This is not a clear discussion of the results. Please, try to provide a more detailed hypothesis.

L340-348: Perhaps it should be better referring to a correlation analysis rather than to a GLM that is the procedure used for testing the correlation.

CONCLUSIONS

L362-368: this is a summary of the results.

L368-371: honestly this conclusion seems very generic and not fully supported by the results since each system showed strengths and weaknesses.

Reviewer #2: Review of the manuscript “Behaviour and Welfare Assessment of Autochthonous Slow-Growing Rabbits: The Role of Housing Systems”

Broad comments:

The authors are based in a country where rabbit production is very important. Thus, this subject is relevant for the field and address a welfare issue, which are housing systems, which greatly affect rabbits’ welfare. Behavioral indicators were well evaluated and described, including the studied repertoire and their approach. However, the corticosterone evaluation seems not to be adequate for the specie. There is little information on literature regard their corticosterone circadian rhythm to state that the studied animals were more stressed based in this single measure. Moreover, just a single-time measure was performed, which does not give us an overview of the circadian rhythm. I am not convinced that the hair corticosterone is informing something. Based in other reviewers’ comments, may it would a good idea to remove your corticosterone data from your manuscript and describe and explore more your behavioral data, despite the fact that you compare single housing with group housing for a social animal. In a “perfect” world, you must have a “perfect” environment, and explore and measure how an environment with more space, ground contact, and complex social network may affect the individuals.

Moreover, there are important information missing. Please, find my specific comments bellow.

Specific comments:

Line 59-60: Please, insert reference regarding aggressive behavior.

Line 70: The ethogram is just a tool.

Line 79-86: Please, standardized the measure units (mm, m2, animals per square meter or <40kg per m², etc). In order to make more comparable, I suggest standardize for kg/m²).

Line 89: “+15/+28 °C” Do you mean 15-25°C?

Line 91: ad libitum

Line 97: used instead of employed.

Line 98: How did you define these ages for behavioral measures? You must explain the reasons to choose these specific ages. Same comments for the time of the day (9-11 a.m.; 2-4 p.m.).

Line 99: Are rabbits inactive during nighttime?

Line 110: The authors’ name or reference number is wrong. Number 3 reference is Munari et al., and Mugnai et al is reference number 6.

Line 112: That means that you measure only the frequency of the behaviors and not the duration of it. Am I right? If so, please make it clear in the manuscript.

Line 114: This is the first time in the manuscript that is presented this test. There is none description or mention about how this test could help the authors regarding their research question. What time in the day this test was carried out? Prior or previous behavioral evaluation? Was the operator who performed the test the same as the one who conducted the behavioral evaluation? This is an important information, since the presence of the person who did the test may interfere on rabbits’ behavior.

Line 119: Are you sure this is the correct reference? The cited manuscript did an open-field test and did not mention any immobility test. Moreover, I did not find any information regarding this test in the manuscript. The authors should include why you performed this test and how these data could contribute to your research question.

Line 141: what time in the day did you collect saliva? It was early morning, late morning, early afternoon. This information is crucial to understand the circadian rhythm of corticosterone.

Line 149-166: I am concern about your measurement in hair cortisol. The protocol seems adequate; however, the matrix is not the most indicate. We have scientific evidence in other species that the hair concentration can differ depending on the color of the hair in dogs (Benett and Hayssen, 2010); age and color (Binz et al., 2018); body regions and color in cows (Burnett et al., 2014); body regions, age, hair color, and season in cows and in pigs (Heimbürge et al., 2020), among others. Moreover, your reference number 17 and 18 in the text are different from your reference list. You cited Meyer et al as 17 but in the reference list is number 18. Please, take a thorough look in all your references links.

Line 232: Could you mention here examples of “potential negative effects associated with sexual maturity”

Line 267-268: I did not understand this statement here since you did not provided an environmental enrichment for the rabbits.

Line 288: “…because rabbits are a social species”

Line 357: here is the major example of how your stress data are questionable. How a social animal, selected from generations and generations to be social, could be more stressful when group housed?

6. PLOS authors have the option to publish the peer review history of their article (what does this mean?). If published, this will include your full peer review and any attached files.

Reviewer #1: No

Reviewer #2: **Yes: **Thiago Bernardino

---

## [Author Response · Author response to Decision Letter 0]

2 May 2024

Response to Editor: 

Dear Dr. Portugaliza

Thank you for considering our manuscript. We appreciate the opportunity to revise and improve our work to better meet PLOS ONE's publication criteria. We will carefully review the feedback provided and make the necessary revisions accordingly. We are committed to addressing the points raised during the review process to enhance the quality and relevance of our research. We look forward to resubmitting the revised manuscript and hope that it will meet your expectations.

Regarding the additional requirements: 

1- We apologize for any oversight regarding PLOS ONE's style requirements, including file naming conventions. We have reviewed and adjusted our manuscript to ensure compliance with these guidelines. Our aim is to adhere to all necessary formatting standards to facilitate a smooth review process. We appreciate your guidance and will ensure that future submissions align closely with PLOS ONE's style requirements.

2- In this trial, our methodology involved raising animals until they reached the appropriate age for slaughter, without directly conducting the slaughter process. Consequently, we lack specific information regarding (1) methods of sacrifice, (2) anesthesia and/or analgesia techniques employed, and (3) efforts undertaken to alleviate any potential suffering experienced by the animals.

3- The role the funders took in the study is now clearly stated as you indicated.

4- Upon acceptance of the paper, we are fully committed to providing the data in a public repository such as Zenodo, as requested. We recognize the importance of transparency and scientific rigor in research, and we are dedicated to ensuring that our data is readily accessible to the scientific community for verification and further analysis. Thank you for emphasizing the significance of this practice, and we assure you that we will comply with this requirement without any hesitation.

5- We have amended our manuscript to ensure that the ethics statement is appropriately located within the Methods section as per the journal's requirements. 

6- Thank you for bringing this matter to our attention. We sincerely apologize for the oversight regarding the reference to the figure in our manuscript. Upon review, we have indeed removed the figure in question and have made the necessary adjustments in the text accordingly.

7- Captions for the Supporting Information files have been appended at the end of the manuscript, ensuring clarity and proper referencing.

Reviewer #1

GENERAL COMMENT

The present work was aimed at evaluating the behaviour and some physiological indicators of stress in two Italian local breeds of rabbits reared in conventional and alternative housing systems. The topic of the present work is interesting, but the manuscript needs a revision to clarify some key points.

In my opinion the work would have been more impactful with the presence of a commercial hybrids used as control. This comparative approach would allow for a better understanding of how different genotypes respond to challenging conditions and could contribute to discussions on the sustainability and resilience of various livestock farming systems. It may be beneficial for future research to consider such comparisons to enhance the overall impact and relevance of the findings.

The study could benefit from additional recordings to offer a more comprehensive evaluation of the impact of housing systems on animal welfare. Specifically, the absence of lesion scoring is a notable limitation, as it hinders a thorough assessment of potential physical impacts on the animals. Furthermore, including basic data on live weight, growth rate, and feed efficiency would enhance the overall understanding of the animals' physiological responses and performance under different housing conditions. Incorporating these additional measures would contribute valuable insights to the study, providing a more holistic perspective on the welfare and productivity implications of diverse housing systems.

One notable limitation of this paper pertains to certain gaps in the materials and methods section. Specifically, there is a need for clarification regarding key information, such as the number of replicates per group, the total number of observed rabbits, and the sex of the animals. These details are crucial for a comprehensive understanding of the experimental design and the robustness of the study’s findings. Addressing these omissions would enhance the transparency and reliability of the research, providing readers and fellow researchers with essential insights into the experimental setup and the transferability of the results.

Response: 

Thank you for your thoughtful review and valuable feedback on our manuscript. We appreciate your recognition of the importance of the topic and your suggestions for improving the clarity and impact of our work.

Your suggestion regarding the inclusion of a commercial hybrid breed as a control group is indeed valuable. We agree that such a comparative approach would offer valuable insights into how different genotypes respond to various housing systems, but our goal was to focus on giving specific information about these two rabbit populations, investigating if there are differences in behavior and adaptability or if these animals can be assimilated as a whole.

We also acknowledge your point regarding the absence of lesion scoring which might be a limitation. 

We acknowledge the significance of incorporating data on live weight, growth rate, and feed efficiency to gain a deeper understanding of the physiological responses and performance of rabbits across different housing conditions. However, our study aimed to concentrate on observing the behavioral and welfare alterations exerted on these animals by the farming system. Undoubtedly, a future study, perhaps including a commercial control, focused on growth performance will be our priority. Furthermore, we value your feedback concerning gaps in the materials and methods section. We acknowledge the importance of providing clear and detailed information regarding experimental design and parameters. Consequently, we have revised the manuscript to address these omissions, ensuring transparency and reliability in our reporting.

Overall, your feedback will greatly contribute to improving the quality and impact of our research. We are committed to addressing your suggestions and enhancing the rigor and relevance of our work. Thank you once again for your valuable input.

SPECIFIC COMMENTS

ABSTRACT

L13: from this line the authors introduce the term “populations”. It should be clarified if these are two populations of the same breed or if they are two different breeds.

Response: 

The term "population" has been chosen to differentiate the animals because, despite these rabbits not being officially recognized as a breed, they exhibit unique characteristics and are widely spread and recognized in Italy, having a strong connection with the territory. To clarify the utilization of the term 'population' mentioned herein, we have added this sentence in the abstract: “These rabbits are not yet officially recognized as breeds, but they are commonly used in Italy for meat production and represent a distinctive phenotype and local heritage among farmers and consumers.”

L13: The authors could introduce the acronym for the two populations.

Response: 

We did not introduce acronyms in the abstract, following the guidelines of the journal, which suggest avoiding their use in this section.

L16: before the results it would be useful to report some other information regarding materials and methods, i.e., number of animals and replicates per group, duration of the trial, age of the animals.

Response:

Thank you for your feedback. We appreciate your suggestion regarding the inclusion of additional information. We have incorporated details such as the number of animals and replicates per group, the duration of the trial, and the age of the animals before presenting the results. This will ensure clarity and completeness in our methodology description.

L25: did the rabbits reach the sexual maturity during the trial?

Response: 

Yes, the rabbits reached sexual maturity during the trial, and their age was specified in the text.

L26-28: is this the first study that demonstrate the correlation between agonistic behaviours and corticosterone levels?

Response:

The correlation between corticosterone levels and agonistic behaviors has been well established since the study by Girolami et al. (1996). We have added this reference in the discussion section and in the references list.

INTRODUCTION

L41-43: this sentence does not seem particularly useful. It could be cancelled.

Response: 

We have removed this sentence as suggested.

L54-55: there is recent scientific literature regarding the characteristics of alternative rearing systems including information on cage/pen dimensions and environmental enrichment for group-housed rabbits. It could be useful to include a brief description and cite pertinent articles.

Response: 

Thank you for bringing attention to the recent scientific literature on alternative rearing systems for rabbits. It's indeed crucial to consider factors such as cage/pen dimensions and environmental enrichment when discussing the welfare of group-housed rabbits. We have included a brief overview of these findings in this section providing citations to support our points. This will help us better understand the optimal conditions for rabbit rearing and enhance the quality of our work. We have added the following sentences in the introduction, and we have cited the pertinent articles as suggested by the reviewer: “Alternative rearing systems for rabbits encompass a variety of approaches designed to improve animal welfare and optimize production efficiency. Central to these systems are considerations such as cage/pen dimensions and environmental enrichment, which play pivotal roles in promoting the physical and psychological well-being of group-housed rabbits [5]. Cage or pen dimensions in alternative systems often prioritize spaciousness to allow for increased mobility and social interaction among rabbits [6]. Additionally, environmental enrichment strategies such as providing tunnels, platforms, or chew toys offer opportunities for mental stimulation, physical exercise, and natural behaviors such as burrowing and exploring. These elements are essential for ensuring the overall health and happiness of group-housed rabbits, contributing to their quality of life and productivity in alternative rearing systems [7].”

L63-65: it is clear, but why did you choose two populations of the same breed? It should be motivated in the introduction because in terms of impact it would be more useful to compare two different local breeds or a local breed with a commercia genotype. This is what I would have expected reading this sentence.

Response:

Thank you for your input. We selected these two populations because they are representative of the local territory, and our aim was to identify potential differences between them in relation to the type of farming system. While comparing different local breeds or a local breed with a commercial genotype could offer valuable insights, our focus was specifically on assessing variations within these specific populations under different breeding conditions. We will consider your suggestion for providing additional context in the introduction to clarify our rationale for this choice.

L67: it could be useful to introduce the name of the two populations and their acronyms here.

Response: 

Thanks for this suggestion. We have added the names and the acronyms for the two populations as suggested.

MATERIALS AND METHODS

L77-86: It is necessary to provide clear information regarding the number of replicates per group (cages/pens), and the sex of the animals.

Response:

Thank you for this suggestion. To provide clarity on the methodology, we have incorporated all the requested information from the reviewer into Section 2.1 'Animals and Housing'.

L79-83: For the housing systems I would use acronyms that can be easily remembered and interpreted, e.g., Single, Group, Mixed.

Response:

We fully agree with the reviewer that 'Single,' 'Group,' 'Mixed' are acronyms that can be easily remembered and interpreted. Therefore, from this point forward, we have replaced the acronyms 'S,' 'G,' 'C' with 'Single,' 'Group,' 'Mixed'.

L91: What exactly was the slaughter age?

Response: 

We have specified the exact slaughter age as suggested.

L91-93: Since the authors monitored the health (I think morbidity and mortality) I would expect a description of these results in the next section (Results).

Response:

We conducted regular monitoring of various parameters throughout the trial to ensure the well-being of the animals involved, with the primary goal of promptly identifying and removing any potentially sick or distressed individuals to prevent unnecessary suffering. Notably, no sick rabbits were identified during the course of the trial.

A significant consideration in our experimental design was the decision to conclude welfare evaluations at 100 days of age. This decision was made proactively to mitigate the potential need to remove group animals due to aggressive behaviors, particularly as rabbits approached sexual maturity. While aggressive behaviors observed before the 100-day mark were not detrimental to the rabbits' welfare, we anticipated that such behaviors might escalate in severity post-sexual maturity. Consequently, continuation of the trial beyond this point could have necessitated the removal of group animals, which would have compromised the integrity of the study.

By focusing our investigation on the impact of reaching sexual maturity on rabbit behavior while ensuring their well-being and minimizing distress, we aimed to provide a comprehensive understanding of this aspect without subjecting the animals to undue harm or disruption. We believe that this rationale is now more clearly articulated and integrated into the conclusions of our study.

L96-97: Please specify the exact number of rabbits observed per group.

Response: 

We have specified the exact number of rabbits observed per group as suggested.

L98-100: This is a weakness of the work since it is widely known that rabbits are active during night. The authors should consider this aspect when discussing the results.

Response: 

Thank you for your comments, which have enabled us to clarify the circadian rhythm of the observed rabbits. To address this point, we have included the following sentences in the discussion section:

“Our study may have limitations considering that observations of animal behaviors were conducted during daylight hours. European wild rabbits are primarily active at night, particularly at dusk and dawn, and they typically reside and rest during the daytime in their warrens (Villafuerte et al., 1993). However, it has been demonstrated that external factors such as noise or scheduled feeding during the light period can shift their activity patterns, making them predominantly diurnal (Diez et al., 2013). Thus, our observations remain reliable regarding the behavior of these animals, and our results can be generalized for rabbits held in captive conditions”.

L175-184: Did you perform a check of data distribution? Which distribution was set for the model?

Response: 

Yes, we performed a check of data distribution. Normality of data distribution was assessed using the Shapiro-Wilk test. For the Generalized Linear Model (GLM) used to explore the relationships between the rabbits' behaviors and CORT levels, we assumed a gamma distribution with a log link function. This distribution was chosen based on its suitability for modeling non-normally distributed continuous data. The log link function was selected to ensure that the model predictions remained on the positive scale, consistent with the nature of CORT data. We have added this information in section 2.6 “Statistical analysis”.

RESULTS

L195: The acronyms should appear in the abstract and then the first time you define the population in the main 

---

## [Decision Letter · Decision Letter 1]

17 Jun 2024

PONE-D-24-03037R1Behaviour and Welfare Assessment of Autochthonous Slow-Growing Rabbits: The Role of Housing SystemsPLOS ONE

Dear Dr. Fiorilla,

Thank you for submitting your manuscript to PLOS ONE. After careful consideration, we feel that it has merit but does not fully meet PLOS ONE’s publication criteria as it currently stands. Therefore, we invite you to submit a revised version of the manuscript that addresses the points raised during the review process.

**ACADEMIC EDITOR: **Please address the comments of the reviewer, particularly about the corticosterone data, environmental enrichment, and the concept that group-housed rabbits were “more stressed” than caged animals. 

We look forward to receiving your revised manuscript.

Kind regards,

Harvie P. Portugaliza, D.V.M., Ph.D.

Academic Editor

PLOS ONE

Journal Requirements:

Reviewers' comments:

Reviewer's Responses to Questions

**Comments to the Author**

1. If the authors have adequately addressed your comments raised in a previous round of review and you feel that this manuscript is now acceptable for publication, you may indicate that here to bypass the “Comments to the Author” section, enter your conflict of interest statement in the “Confidential to Editor” section, and submit your "Accept" recommendation.

Reviewer #2: (No Response)

2. Is the manuscript technically sound, and do the data support the conclusions?

Reviewer #2: Partly

3. Has the statistical analysis been performed appropriately and rigorously? 

Reviewer #2: Yes

4. Have the authors made all data underlying the findings in their manuscript fully available?

Reviewer #2: Yes

5. Is the manuscript presented in an intelligible fashion and written in standard English?

Reviewer #2: Yes

6. Review Comments to the Author

Reviewer #2: Broad comments:

The authors made a great job, answering every single comment made by both reviewers. However, some specific comments needs to be made.

My main concern is regarding your corticosterone data. As I previously said, you performed a single measured per day and you answered that you performed a repeated measures over time (T1, T2, T3, and T4). However, I was referring to the circadian rhythm! A single information per day is not enough to state that the animal is stressed. I completely understand that you sampled the animals in different days, but since rabbits have nocturnal habit (you mention it in line 356) this will certainly influence their salivary corticosterone concentration over the day hours. Moreover, your explanation regarding rabbits kept in captive conditions and being predominantly diurnal is very valuable! However, the circadian rhythm will follow as a diurnal animal (pig, cattle, poultry, etc), being high in the morning and low in the afternoon. Thus, to state that the animal is stressed you MUST measure this variation or ratio. I understand how complex and time consuming is to obtain these data, but you must inform you reader the limitations on your corticosterone data or remove it. About hair cortisol, the fact that the studied rabbit population have an uniform hair coat color is good but is not a prove that there is a constant and standardized deposition of corticosterone in the hair.

Regarding your answer about my previous comment about environmental enrichment. I understand the impact of housing a social animal in a group and all social benefits compared with individual or crated animal and its impact on welfare. However, if you housed a social animal in group housing and did not provide any environment enrichment their behavioral needs it will not be met, since the environment would be very different than the environment faced in their “habitat”. Of course, that in comparison with single housed or crated, group housing will lead to a better welfare.

My last question was regarding the finding of the manuscript that group-housed rabbits were “more stressed” than caged animals. I completely understand that those animals were living in an artificial condition. Nevertheless, for how many generations? They went through “natural” conditions for much more time and countless generations than in artificial conditions. Did they physiology change over few artificial raised generations?

Your explanation was very plausible, and I agree with your statement. However, I did not believe that your data support the fact that group-housed animals are more stressed (see my comments above about corticosterone) based in limited housed conditions. You did a great job in your discussion, which I believe it is much better written and with more clear and transparent information.

Specific comments:

Line 17-18: “… 294 weaned males with 35 days old were divided…”

Line 64: “… and psychological welfare of group-housed rabbits.”

Line 101-110: Please, standardize you units. Density for Traditional single cage is expressed as animals per square meter and Group farming is expressed as kg per square meter. Mixed pilot systems have both units, kg/m² and animals/m². I already asked this during my first review.

7. PLOS authors have the option to publish the peer review history of their article (what does this mean?). If published, this will include your full peer review and any attached files.

Reviewer #2: **Yes: **Thiago Bernardino

---

## [Author Response · Author response to Decision Letter 1]

19 Jun 2024

Reviewer #2: Broad comments:

Thank you very much for your positive feedback and for acknowledging our efforts in addressing the comments. We appreciate your thorough review and are grateful for the opportunity to improve our work. 

We would like to address your question by explaining our choice of measuring corticosterone in both saliva and hair. We understand your concern about the importance of circadian rhythms, which is why we decided to focus on both saliva and hair corticosterone. Our approach involves collecting and studying saliva corticosterone to provide measurements of acute stress. Since we always collected samples at the same time in the morning, this measurement reflects the stress from the previous night when the rabbits are more active.

We acknowledge that our sampling may not perfectly reflect circadian variations. However, we believe that this method could provide valuable indications of acute stress, and, by studying hair corticosterone, we can also investigate chronic stress. Repeated measurements over time (T1, T2, T3, and T4) allow us to evaluate the progression of stress levels in these animals. Consequently, in line with your comments, we have included the limitations regarding circadian rhythms in the discussion section of our manuscript to clarify our methodological choices and discuss the relevant constraints.

“One limitation of our study is that our sampling method does not fully account for the circadian variations in corticosterone levels. Corticosterone levels naturally fluctuate throughout the day, and our fixed sampling time may not capture these dynamic changes accurately. As a result, while our morning saliva samples reflect the stress levels from the preceding night, they may not provide a comprehensive view of the fluctuations in corticosterone that occur over a 24-hour period.”

Regarding the concern about uniform hair coat color, we agree that while it is beneficial, it does not ensure a constant and standardized deposition of corticosterone in the hair. We have included these limitations in the discussion section of our manuscript.

“We acknowledge that a limitation of using corticosterone in hair as a biomarker for chronic stress is the individual differences in hair coloration. Even within a population of rabbits with uniform hair color, there may be individual variations that require further investigation.”

Regarding environmental enrichment for social animals in group housing, it is crucial to address the need for appropriate environmental stimuli to meet behavioral needs, particularly when animals are housed together. In Europe, there is a growing movement towards abolishing single cages in favor of group housing for social animals. This shift reflects increasing recognition that group housing constitutes a substantial form of environmental enrichment, as recognized by both national and European regulations. For example, the European Union Directive 2008/120/EC on the protection of rabbits specifically mandates that rabbits must be housed in compatible groups unless specific reasons justify individual housing. This regulation acknowledges that group housing allows rabbits to engage in natural social behaviors, promoting better welfare compared to solitary or crated housing. This approach is in line with evolving regulatory frameworks and societal expectations aimed at advancing animal welfare standards across Europe.

Finally, regarding the last suggestion, the welfare challenges observed in rabbits raised in group housing settings primarily stem not from genetic factors but from the type of farming system itself, which may not be well-suited for rabbits, especially robust breeds like the Grey rabbits of Piedmont. This indigenous Italian breed has historically been raised in both cage-based and more extensive farming systems, known for their resilience and adaptation to local conditions rather than intensive cage farming. Our research focuses on addressing increasing consumer demand and European Union regulations aimed at phasing out cage systems in animal farming. These rabbits naturally exhibit behaviors such as dominance and aggression among males, complicating their transition to group housing arrangements. To mitigate these challenges, we propose a two-phase farming approach detailed in our paper. This approach considers their natural social dynamics and temperament, aiming to ensure their well-being while meeting market demands. Furthermore, due to their medium to slow growth rate compared to intensive hybrid breeds, the Grey rabbits of Piedmont require longer farming periods. This emphasizes the importance of adopting sustainable and welfare-oriented farming practices that cater specifically to their unique needs and behaviors.

Specific comments:

Line 17-18: “… 294 weaned males with 35 days old were divided…

Corrected as suggested

Line 64: “… and psychological welfare of group-housed rabbits.”

Corrected as suggested

Line 101-110: Please, standardize you units. Density for Traditional single cage is expressed as animals per square meter and Group farming is expressed as kg per square meter. Mixed pilot systems have both units, kg/m² and animals/m². I already asked this during my first review.

Thank you for your feedback. We have made the necessary corrections, and now all density values are uniformly expressed in kg/m² as requested.

---

## [Editor Report · Decision Letter 2]

5 Jul 2024

Behaviour and Welfare Assessment of Autochthonous Slow-Growing Rabbits: The Role of Housing Systems

PONE-D-24-03037R2

Dear Dr. Fiorilla,

We’re pleased to inform you that your manuscript has been judged scientifically suitable for publication and will be formally accepted for publication once it meets all outstanding technical requirements.

Kind regards,

Harvie P. Portugaliza, D.V.M., Ph.D.

Academic Editor

PLOS ONE
---

## [Editor Report · Acceptance letter]

10 Jul 2024

PONE-D-24-03037R2 

PLOS ONE

Dear Dr. Fiorilla, 

I'm pleased to inform you that your manuscript has been deemed suitable for publication in PLOS ONE. Congratulations! Your manuscript is now being handed over to our production team.

Kind regards, 

on behalf of

Dr. Harvie P. Portugaliza 

Academic Editor

PLOS ONE